# Deep Learning and Statistical Models for Forecasting Transportation Demand: A Case Study of Multiple Distribution Centers

Fábio Polola Mamede [1,*], Roberto Fray da Silva [2], Irineu de Brito Junior [1,3], Hugo Tsugunobu Yoshida Yoshizaki [1,4], Celso Mitsuo Hino [4] and Carlos Eduardo Cugnasca [1]

1 Graduate Program in Logistics Systems Engineering, University of São Paulo, São Paulo 05508-010, Brazil; irineu.brito@unesp.br (I.d.B.J.); hugo@usp.br (H.T.Y.Y.); carlos.cugnasca@usp.br (C.E.C.)
2 Institute of Advanced Studies, University of São Paulo, São Paulo 05508-010, Brazil; roberto.fray.silva@usp.br
3 Environmental Engineering Department, São Paulo State University, São José dos Campos 12247-004, Brazil
4 Department of Production Engineering, University of São Paulo, São Paulo 05508-010, Brazil; cmhino@usp.br
* Correspondence: fabiomamede@alumni.usp.br

**Abstract:** *Background*: Transportation demand forecasting is an essential activity for logistics operators and carriers. It leverages business operation decisions, infrastructure, management, and resource planning activities. Since 2015, there has been an increase in the use of deep learning models in this domain. However, there is a gap in works comparing traditional statistics and deep learning models for transportation demand forecasts. This work aimed to perform a case study of aggregated transportation demand forecasts in 54 distribution centers of a Brazilian carrier. *Methods*: A computational simulation and case study methods were applied, exploring the characteristics of the datasets through autoregressive integrated moving average (ARIMA) and its variations, in addition to a deep neural network, long short-term memory, known as LSTM. Eight scenarios were explored while considering different data preprocessing methods and evaluating how outliers, training and testing dataset splits during cross-validation, and the relevant hyperparameters of each model can affect the demand forecast. *Results*: The long short-term memory networks were observed to outperform the statistical methods in ninety-four percent of the dispatching units over the evaluated scenarios, while the autoregressive integrated moving average modeled the remaining five percent. *Conclusions*: This work found that forecasting transportation demands can address practical issues in supply chains, specially resource planning management.

**Keywords:** transportation demand forecasting; supply chain management; LSTM; ARIMA; data preprocessing





## 1. Introduction

Logistics providers and shipping companies need to define and organize their cross-docking operations and warehouses to orchestrate product consolidation and distribution to their customers. A logistics provider can carry out shipments in different sectors, such as supplies [1], wholesales [2], retailers [3], industry, e-commerce, and others. One significant factor influencing transportation demands is the order cycle [4], as each delivery depends on the customer's demand [5]. Additionally, uncertainty plays a significant role in transportation demand in areas such as sales [1], manufacturing [3], inventory [6], organizational risks, lead times [7], and the economy [8], making it challenging to make predictions. However, improving the accuracy of transportation demand prediction could significantly improve the decision-making processes of different agents in the supply chain.

A transportation demand forecast can be defined as the number of orders in terms of weight, volume, or quantity of products shipped in a period. It is meaningful for resource

planning [9], such as planning equipment availability in a location. Forecasting transportation demands in advance can contribute to finding more operational opportunities and supporting the logistics providers' decisions [5,6].

The state-of-the-art literature to predict transportation demands uses machine learning and deep learning methods [9]. Nevertheless, traditional statistical models are also used for their satisfactory results in predicting different time series. Ref. [7] observed that the most commonly used methods for supply chain demand forecasting methods are artificial neural networks, regression models, statistics methods, support vector machines (SVM), and decision trees.

The literature has a few works on transportation demand forecasts; however, each one has a different focus, such as the logistics service capacity [6], business volume [10], or scheduling [11]. None of these works or other references have applied transportation demand forecasts to predict resource availability in facilities. Therefore, there is a gap in the state-of-art literature as regards exploring the business case of repositioning vehicles over facilities.

The work by [10] evaluated the use of deep neural networks in this domain, identifying their potential applications in real-world scenarios. The works by [6,8,12,13] explored different aspects of using machine learning methods to predict transportation demands. There seems to be a consensus in the literature that using machine learning methods, especially deep learning, may improve the accuracy of transportation demand prediction.

This work aims to analyze and compare the use of autoregressive integrated moving average (ARIMA) models and their seasonal variation (SARIMA) and long short-term memory (LSTM) neural networks to forecast transportation demands at the warehouses and cross-dock locations of the logistics provider, addressing the gap mentioned. The main goals for this work are (1) to carry out an exploratory data analysis to understand the data characteristics; (2) to implement ARIMA and LSTM models with their key hyperparameters; and (3) to evaluate the results, verifying the forecasts' accuracy and processing time.

Various scenarios are analyzed, considering different data preprocessing and outlier treatment methods. The methodology used can be applied to other cases and scenarios, and the results could better inform decision-makers in different supply chain links. Additionally, it should be noted that our results also provide an in-depth comparison of traditional statistics and deep learning models, thus addressing a significant gap in the literature on transportation demand prediction or forecasting.

The following research questions are evaluated in this research: (i) Which of the models evaluated presents the best results regarding lower average error metric values? (ii) Is the behavior of the models equivalent for all shipping units? (iii) How does data treatment influence forecast results? and (iv) How did the COVID-19 pandemic influence predictions for different distribution centers and scenarios?

The document is organized as follows. Section 2 presents the theoretical foundations of the models used and of times series analysis and prediction. Section 3 describes the materials and methods used in the case study. Section 4 presents the results of the exploratory data analysis and model comparisons for the different scenarios. Section 5 discusses the potential impacts of this research; Section 6 concludes this work, providing recommendations for future studies.

## 2. Theoretical Foundations

This section describes the theoretical foundations related to functional aspects (Section 2.1); the use of traditional statistics methods, with an emphasis on the ARIMA model (Section 2.2); the use of deep learning models, especially LSTM, which is considered the state-of-the-art for this domain (Section 2.3); and different data preprocessing methods and their relevance (Section 2.4).

### 2.1. Functional Aspects

The ability to predict the future based on historical data is essential to improving decision-making in supply chains [4], thus supporting individual and organizational decision-making. The literature highlights the relevance of transportation demand forecasting, with increasing applications from 2005 onwards [7]. The work by [7] points out that demand forecasting has a time series characteristic [9], defined through a historical basis [10,14]. The work by [9] used artificial neural networks to forecast demands, as these models are suitable for modeling non-linear time series. Another work proposed a composition of dynamic machine learning models and architectures for real cases because they can behave both linearly and non-linearly in different situations [15].

Geographic analyses were also evaluated by [16,17], which correlated transportation demand with the influence of various regions, locations, and communities and discussed how this can be addressed by neural network forecasts. The work showed the importance of clustering the datasets into subsets, split by location. Logistics providers work with operational facilities, which can perform different roles in the transport network, such as cross-docks, hubs, and distribution center (warehouses) [11].

The cross-dock location receives inbound trucks and quickly ships out outbound trucks, and the state-of-art of cross-docking is related to distribution, logistics, scheduling, and vehicle availability. Techniques such as linear and integer programming, non-linear programming, and stochastic programming are the most frequently used optimization approaches for cross-docking [18].

Warehouses store products and supply the initial distribution of a chain, and in the last three decades, they have advanced in terms of technology, organization, and automation [19]. In the operation of cross-docks and warehouses, truck appointments and availability are required to guarantee that the operational process considers all the variables, such as yard management, dock scheduling [20], warehouse management [21], and truck availability.

### 2.2. ARIMA Models and Time Series Prediction

ARIMA is a predictive model based on statistics and econometric methods and is commonly used for time series prediction [6]. Due to its characteristics, it is better suited for predicting linear series [6]. The model's implementation is relatively simple, fast, and provides satisfactory results in several domains and datasets [9]. The literature's main works exploring its use for transportation demand forecasting are [6,8,12].

Although ARIMA models are simple to implement, the predictions generated can provide inaccurate results over non-linear time series, as observed by [15]. Differentiation is needed to apply the model to a specific dataset in several cases. ARIMA models can derive additional components, namely exogenous variables (ARIMAX) and seasonal components (SARIMA), which can extract further information from data to improve forecasting results concerning ARIMA. As it is traditionally used in time series prediction problems, it can provide an interesting benchmark to compare with other prediction models.

ARIMA has three main components [22]: (1) autoregression, or $p$, which is a linear regression over the time series historical data and an autocorrelation interval; (2) the integrated part, or $i$, which is used to eliminate non-stationarity in the data; and (3) the moving average, or $q$, which is the autocorrelation interval, used to calculate the error's moving average. For the SARIMA model, additional seasonal components are included to better capture seasonality in the data. The forecasts are provided by a linear combination of each component with past data, as shown in Equation (1):

$$Y_t = \alpha + \beta_1 Y_{t-1} + \beta_2 Y_{t-2} + ... + \beta_p Y_{t-p} \epsilon_t + \phi_1 \epsilon_{t-1} + \phi_2 \epsilon_{t-2} + ... + \phi_q \epsilon_{t-q} \qquad (1)$$

The hyperparameters used to build ARIMA and SARIMA models can be determined by using either analytic methods, such as autocorrelation and partial autocorrelation, or

by exploring error metrics, such as the Akaike information criterion (AIC) associated with different hyperparameter values [23].

### 2.3. Neural Network for Time Series Prediction

Artificial neural networks [24] are composed of processing elements, called neurons, fed by input variables and computed in a final result by an activation function. Neurons are connected by weighted connections, flowing data according to scale, weights, and input values. Artificial neurons are trained so that the weights are regulated according to the function and neural network model used. Ref. [8] reinforces the use of artificial neural networks to forecast demands applied to the scenario of automotive parts.

The state-of-the-art in time series prediction over big data is the use of artificial neural networks architectures called deep learning architectures, which exist in several frameworks [25], such as feed-forward networks, recurrent networks, Elman networks, and others. Recurrent neural networks are suitable for sequential data, such as time series [26]. There is a wide range of applications for big data analytics in supply chain forecasts [27].

LSTM is a deep artificial neural network [28] architecture designed to address time series problems [22]. It provides substantial benefits in relation to other artificial neural networks when working with long time series data by preventing the exploding gradient and vanishing problems. It is widely used in non-linear time series [9,22,29] and has wide applications in natural language processing and time series prediction and analysis. The literature presents many interesting works that address transportation demand forecasts using LSTMs, such as [6,9,12,13].

LSTM networks have gates to input data, and the logic behind the architecture controls how the data are processed, defining relevance and priority, memorizing the most relevant aspects, and disregarding irrelevant information. The neuron states and hidden layers forward data to the next state. Figure 1 illustrates the architecture of an LSTM neural network.

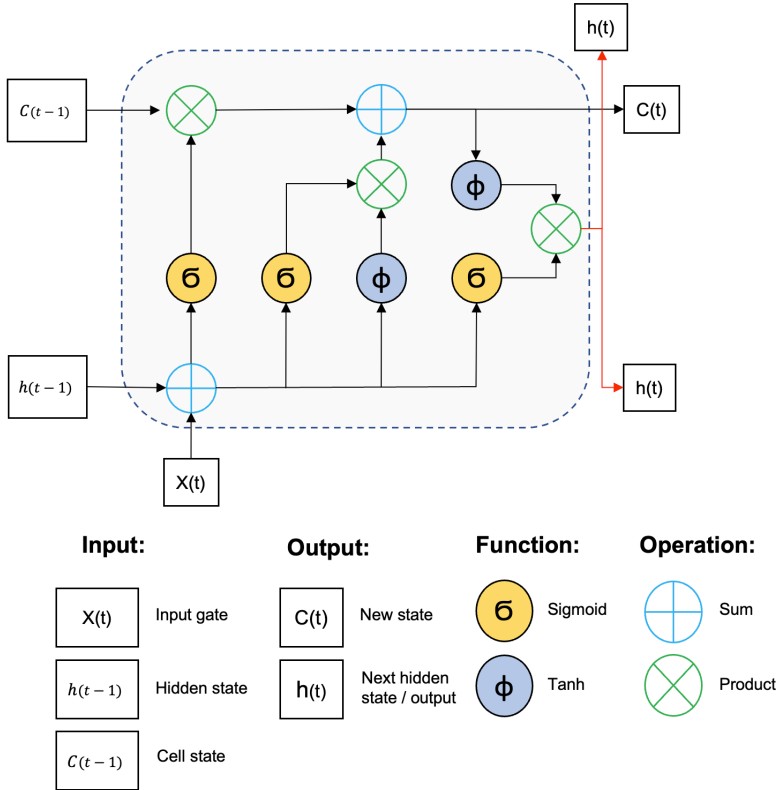

**Figure 1.** LSTM neural network design based on [29].

As an example of the architecture, Figure 2 represents an LSTM neural network designed for a time series with a time step in 7 and 2 layers with 8 neurons each.

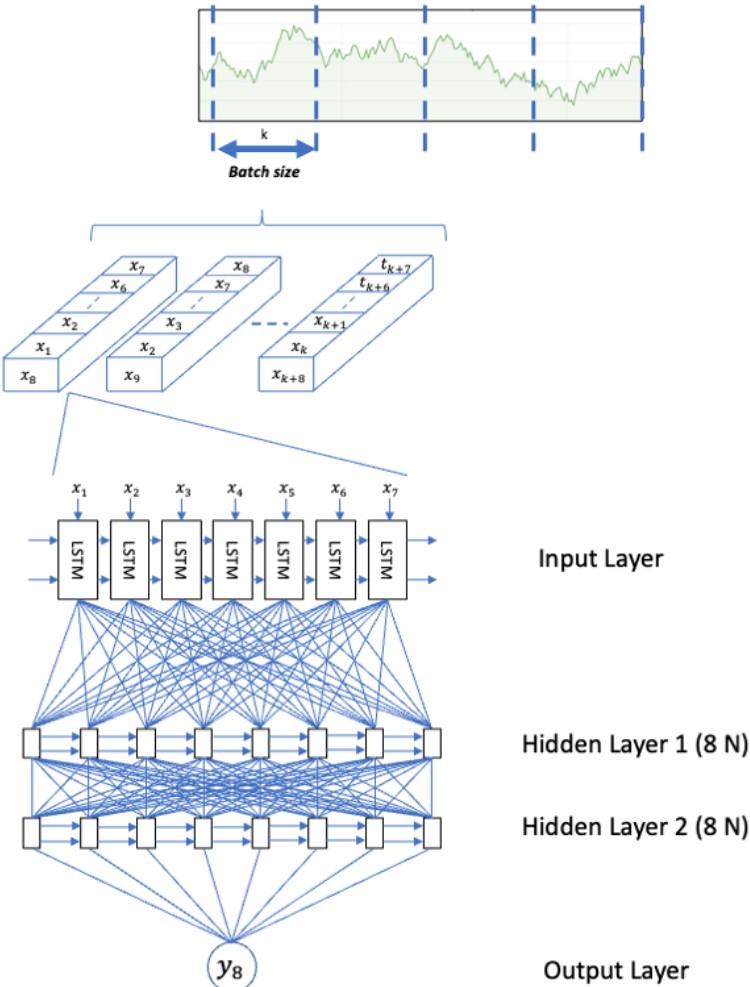

**Figure 2.** LSTM neural network architecture with 2 hidden layers with 8 neurons each.

To model an LSTM, it is necessary to define the values of the hyperparameters composing the model [30]. There are different ways to identify those values, such as charts, heuristics, and combinatory tests. However, no widespread agreement exists on which method can provide the best results [31]. Therefore, the critical hyperparameters explored herein, using a traditional grid search method, were [9,12,14,31–33]:

1.  Lag size (time steps): number of values to define an interval or time window;
2.  Hidden layers: number of neuron layers connected between the input and output layer;
3.  Neurons: total number of neurons in each layer;
4.  Batch size: the size of the batches used before refreshing the weights between the connections;
5.  Epochs: number of iterations repeated to train the model.

## 2.4. Time Series Data Processing: Multiple Methods and Their Uses

The presence of outliers due to randomness or noise in a dataset can increase the prediction complexity [34] and reduce forecast accuracy. Correctly detecting outliers and treating them is essential for specific problems. Their elimination or replacement with treated data (such as the average, median, or mode) can help reduce forecasting errors. Ref. [34] used band filters (minimum and maximum intervals) to identify outliers and

replaced them with the moving average of a predetermined period in addition to using data interpolations. This was essential to improving prediction quality.

Ref. [35] evaluated the impact of data quality on predictions using an LSTM model. The outliers detected were replaced with the median of the variable. Different scenarios were also evaluated considering the comparison with the raw dataset scenario. Ref. [36] replaced the outliers detected in their work using upper and lower limit bands by quantile percentages. Ref. [37] replaced the detected outliers with the median absolute deviation. Lastly, [38] used data interpolation between the nearest data points to replace the outliers detected. All works generally observed that treating outliers is essential to improving prediction accuracy and decreasing error metrics in the test subset.

One of the most recent advances in identifying outliers and anomalies is using computational methods such as the ones used by [39]. Several works have observed that using those techniques, especially one denominated isolation forest, can provide significant benefits for anomaly detection. By using this technique, Ref. [39] observed a 57.5% reduction in the error metrics evaluated in relation to not treating the outliers in their dataset.

## 3. Materials and Methods

This work uses two methods [40], computer simulations [41] and case studies [42]. The first method involves implementing a computational model that allows us to simulate the characteristics of a specific problem, aiming to obtain insights and make predictions about variables or behaviors of interest. The second method is related to applying the implemented model to a real-life situation using data collected from a representative company in the market to distribute products of different natures, origins, and destinations.

The case study represents an actual problem of a logistics provider company that requires forecasting the amount of transport demands to ship products from their warehouses and cross-dock locations. This is then used to improve resource planning and operations. This company has multiple facilities over the Brazilian territory. Each unit can have a warehouse or cross-dock role where the shipments can be consolidated, distributed, or transferred to other facilities or customers. The company owns 800 trucks to transfer products between facilities. Once a truck reaches a unit, it can be reloaded (if there is transportation demand from that specific location) or allocated to another facility to support other locations' demands.

The methodology used has 9 steps, illustrated in Figure 3. The main requirements and conditions for implementation were (first step):

1. Selecting prediction models to be evaluated;
2. Gathering different datasets representing each location and its specific characteristics;
3. Determining the technical feasibility of the models that can be developed or implemented;
4. Ensuring that the forecast models have adequate processing times for application to the current research;
5. Defining the error metrics to be evaluated;
6. Assessing the resilience of the models in different datasets, scenarios, and variations.

In the second step, data were collected from a Brazilian logistics provider. The final dataset contained operational information for 2019 and 2020, with daily transport demands for different products in each of its 54 shipping units. The variables collected were: (1) weight, (2) volume, (3) origin unit, and (4) date of demand. Each shipping unit was given a code composed of U and a sequential number to maintain anonymity. Therefore, 54 datasets were derived from the gathered data, varying from U01 to U54.

Then, in the third step, the datasets were manipulated using Tad [43] and DB Browser for SQLite [44] to view the data, format dates and special characters, and fix incorrect variable types. In the fourth step, the dataset was preprocessed, eliminating non-essential data, grouping, and preparing the dataset for the processing step.

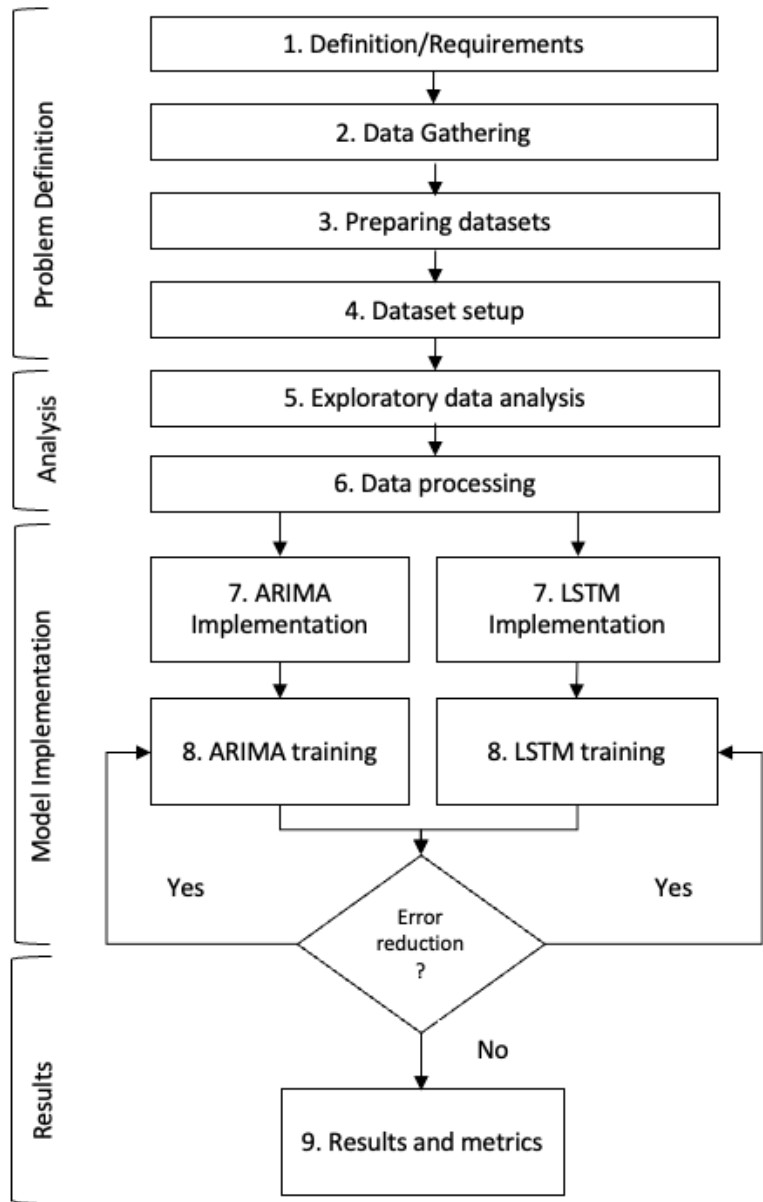

**Figure 3.** Method flowchart.

In the fifth step, an in-depth exploratory data analysis was conducted to extract information about the variable distributions and their most important statistical indices. The sixth step identified potential outliers, and different treatments were applied. Table 1 describes the 8 scenarios evaluated in this work.

Scenarios 1 and 3 evaluate the ARIMA/SARIMA and LSTM models without any outlier treatment. Scenarios 2 and 4 evaluate ARIMA/SARIMA and LSTM while performing outlier treatment, replacing the values identified by the lower or upper limit band. Scenarios 5 and 6 implement ARIMA/SARIMA and LSTM, discarding the identified outliers. Finally, Scenarios 7 and 8 implement ARIMA/SARIMA and LSTM with outlier treatment using advanced techniques for detecting anomalies in the data series. The isolation forest method from the Python library pycaret was used, replacing the points identified by the median of the dataset. All the scenarios considered all the datasets and were built applying the cross-validation technique by dividing the training set into four parts (25%, 50%, 75%, and 100% of the total data in the training subset), as in [45]. Normalization was applied to all variables (transforming them in a range from 0 to 1) so that the model metrics were on the same scale.

**Table 1.** Scenarios selected for the research.

| Scenario | Method | Outlier Treatment |
|---|---|---|
| S1 | ARIMA/SARIMA | No |
| S2 | ARIMA/SARIMA | Lower and upper boundaries |
| S3 | LSTM | No |
| S4 | LSTM | Lower and upper boundaries |
| S5 | ARIMA/SARIMA | Outlier removal |
| S6 | LSTM | Outlier removal |
| S7 | ARIMA/SARIMA | Isolation forest/median |
| S8 | LSTM | Isolation forest/median |

In the seventh step, the models were implemented. In the eighth step, the grid search method was used to generate all the combinations of hyperparameter values for each model, storing the metrics obtained in the training data (first 80% of the total base data) and finally ordering them according to the criteria of smaller MAE metrics. Then, the hyperparameter values with the lowest MAE metric in the training set were selected to be applied to the test set (the last 20% of the total data). The final comparison of the models in the scenarios evaluated was carried out, considering the following metrics: MSE, MAE, and processing time.

This work used the Python programming language and the libraries keras, tensorfow, pandas, matplotlib, statsmodels, scikit-learn, statistics, sklearn, numpy, pycaret, and seaborn.

## 4. Results

This section contains the results of this work, divided into subsections. Section 4.1 contains the exploratory data analysis, Section 4.2 contains the model's results, and Section 4.3 discusses the computational performance.

### 4.1. Exploratory Data Analysis

For each dataset explored in this work, the following types of charts were obtained: (i) a chart of the original series (demand data in a shipping unit in two years); (ii) the original series plus the moving averages and the moving standard deviation, which allowed observing data fluctuations over time; (iii) charts with differentiated series (up to the third differentiation) to evaluate the stationarity characteristics of the datasets; (iv) autocorrelation and partial autocorrelation charts to identify the relationship between the datasets and possible recurrences and contributions to the ARIMA/SARIMA models; (v) time series decomposition charts, separating the dataset into 4 components: data, trend, seasonality, and residuals, in order to improve the identification of seasonality, interference of outliers, and trends; and (vi) histograms of the variables to identify the data frequency and distribution.

Each dataset has independent characteristics, which make them unique and distinct. The charts in Figure 4 corroborate the heterogeneity of the datasets, which have different scales (number of demands) and variations in different periods. These charts depict the datasets and the statistical characteristics of each unit, containing the moving averages and standard deviations and the number of demands per day in the years 2019 and 2020, without any treatment of outliers.

Shipping unit U38 had the lowest quantity of demands, not exceeding 60 daily demands, while unit U45 presented the highest volume of traded orders, exceeding 17,000 demands in a single day. The U12 unit chart exemplifies a unit that had variations in different periods, with increasing and decreasing demands. Other units, such as U51, showed regularity throughout the sampling period. A few units presented a behavior similar to U19, presenting potential seasonality over the period. Lastly, some shipping units, such as the one shown in Figure 4, have missing data in the period due to operational interruptions or transfers to other locations. A feature observed in all units is days with

zero demand values. This occurs on days when the units are operationally inactive, such as weekends or holidays.

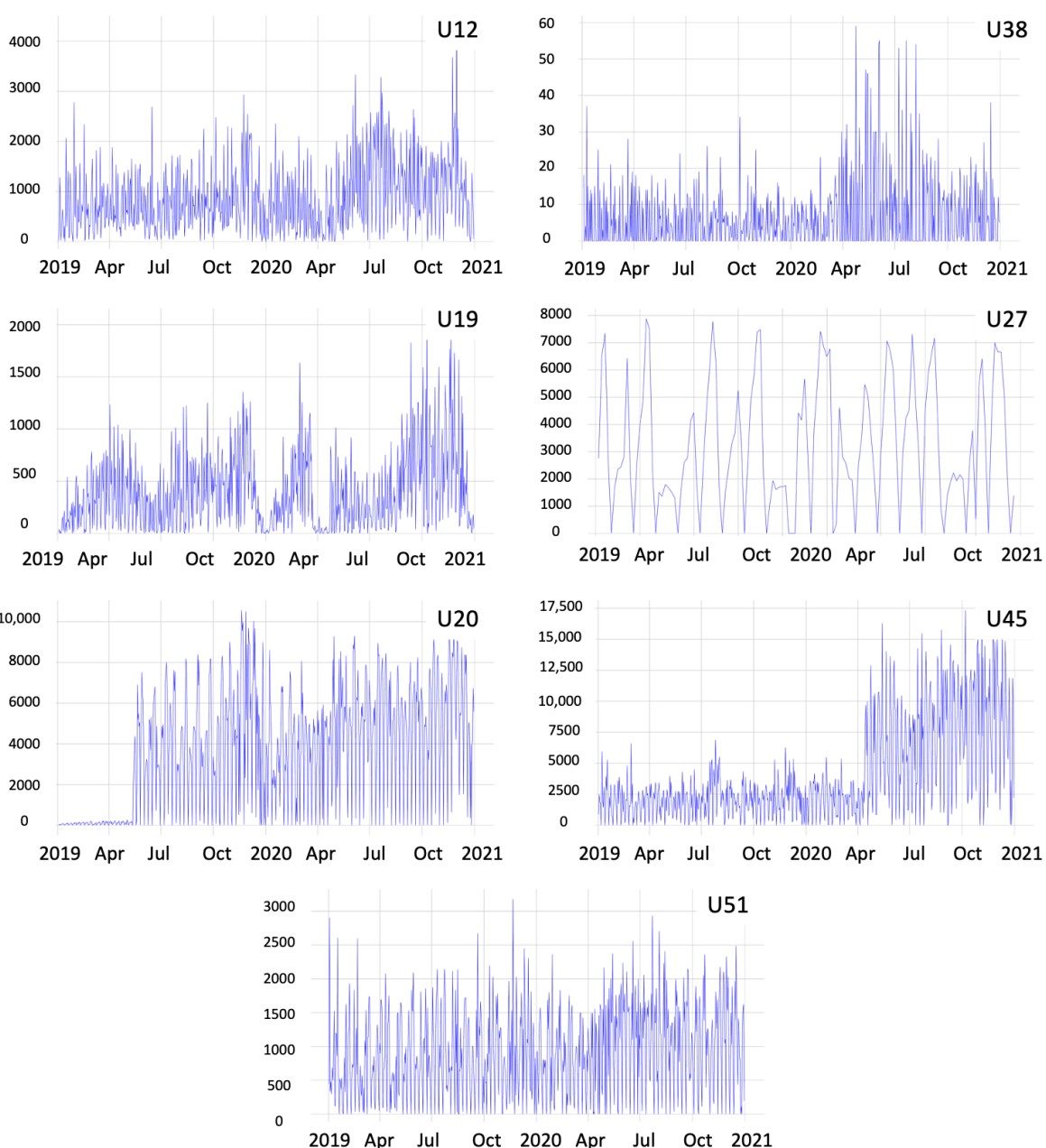

**Figure 4.** Demands of shipping units U12, U38, U19, U27, U20, U45, and U51 with their respective moving averages.

The augmented Dickey-Fuller test (ADF) was verified to assess the datasets' stationarity. Considering the original data (without outlier treatment), approximately 80% of the units presented non-stationary behavior. Those series needed to be differentiated for the ARIMA/SARIMA models to be applied. Figure 5, which contains the autocorrelation and partial charts for unit U05, shows the original and the differentiated series. These charts were used to identify the values of the ARIMA model hyperparameters and the AIC metric calculated for all units within a set of possible values. These charts also help identify seasonal characteristics.

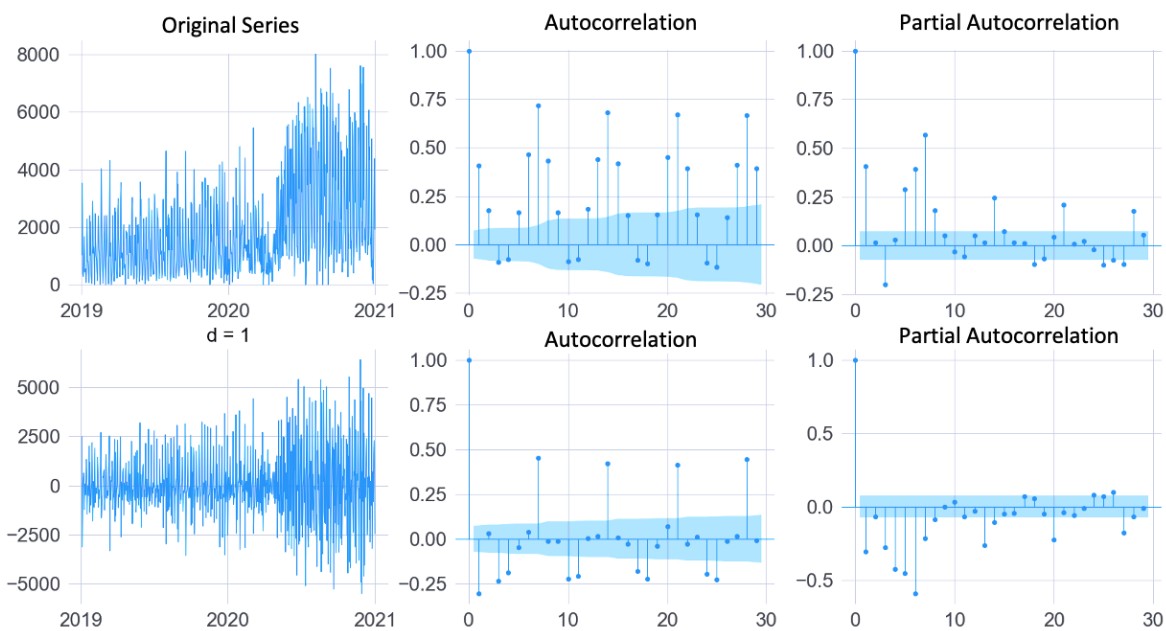

**Figure 5.** Charts of autocorrelated, partially autocorrelated, original, and differentiated series of unit U05.

Other characteristics of the shipping units were obtained from the seasonal decomposition of each dataset, as shown in Figure 6. The data were separated into their base values (A), trends (B), seasonality (C), and residuals (D). In all units, the characteristic of seasonality with weekly recurrence was identified. Another common dataset characteristic was the amplitude and randomness of the residuals, suggesting characteristics such as the presence of outliers and non-stationarity. The trend curve described possible oscillations in specific periods, such as at the end of 2019 and the beginning of 2020, as shown by the unit in Figure 6.

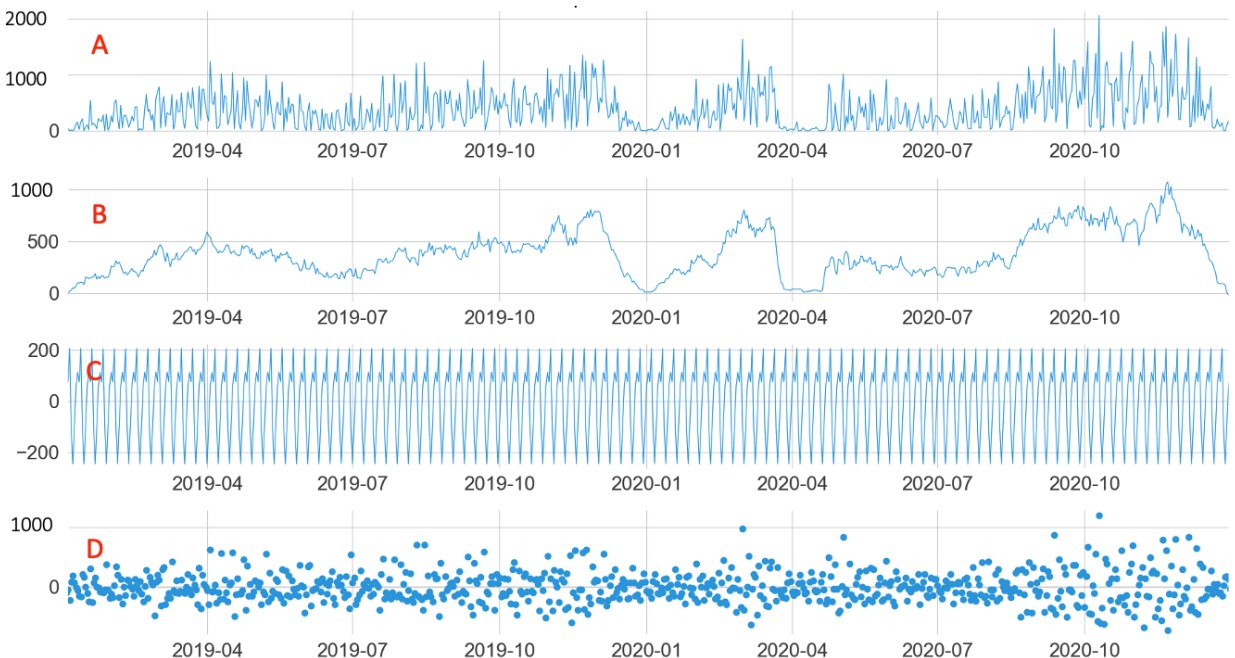

**Figure 6.** Seasonal decomposition ((**A**): original data, (**B**): trend, (**C**): seasonality, and (**D**): residuals) of unit U19. (**A**) + (**B**) + (**C**) + (**D**) represents the demands of the shipping unit time series.

In the period evaluated in this research, which started in December 2019, some signs of the COVID-19 pandemic could be observed, with sanitary restrictions and isolation measures already implemented early in 2020 [46] affecting some areas of the economy [47]. Some shipping units had visible oscillations in 2020, emphasizing the beginning of the year, as seen in Figure 4. Likewise, indications can be observed in some product categories, such as handbags and footwear. Conversely, the IT category (Figure 7) showed an increase in demand after May 2020, which can be interpreted as a result of the increase in home office activities.

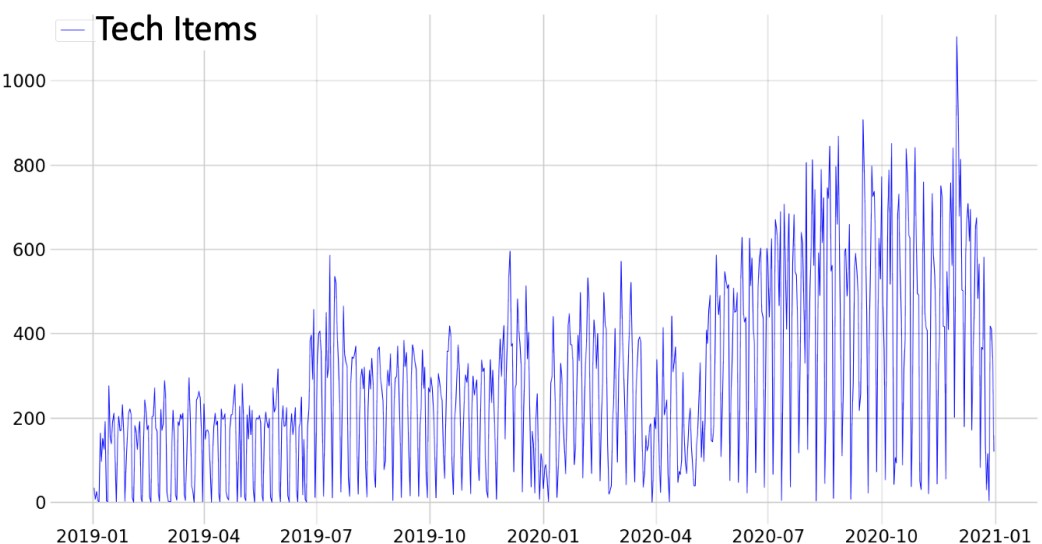

**Figure 7.** Transport demands of the IT categories with their respective moving averages.

### 4.2. Models Results

This subsection presents the results of the ARIMA/SARIMA and LSTM models using the hyperparameter values found in the training subset and applied to the test subset. Table 2 contains the main metrics observed in the test subset for all the units and scenarios. It is essential to observe that some scenarios results in this table contained blank values due to one of the following reasons: (i) original data quality problems or (ii) the amount of data in the dataset was not adequate to allow the models to converge on that specific dataset. The results in bold represent the scenario that obtained the smallest metric, that is, the scenarios for which the predictions reached an accuracy closer to the actual value.

**Table 2.** MAE test metrics for all shipping units and scenarios. The bold numbers represents what is the scenario with smaller error metric in each shipping unit.

| Scenario | S1 | S2 | S3 | S4 | S5 | S6 | S7 | S8 |
|---|---|---|---|---|---|---|---|---|
| U01 | 0.098 | 0.293 | **0.031** | 0.082 | 0.262 | 0.216 | 0.204 | 0.093 |
| U02 | 0.139 | 0.160 | 0.072 | **0.060** | 0.153 | 0.132 | 0.181 | 0.072 |
| U03 | 0.121 | 0.121 | 0.055 | **0.037** | 0.184 | 0.136 | 0.128 | 0.047 |
| U04 | 0.122 | 0.123 | 0.139 | 0.112 | 0.126 | 0.123 | 0.123 | **0.104** |
| U05 | 0.243 | 0.178 | 0.067 | **0.055** | 0.224 | 0.201 | 0.183 | 0.097 |
| U06 | 0.186 | 0.174 | 0.046 | 0.049 | 0.175 | 0.161 | 0.255 | **0.032** |
| U07 | 0.176 | 0.199 | 0.126 | **0.102** | 0.239 | 0.140 | 0.228 | 0.194 |
| U08 | 0.177 | 0.205 | **0.066** | 0.081 | 0.220 | 0.158 | 0.142 | 0.070 |
| U09 | 0.204 | 0.169 | **0.076** | 0.085 | 0.281 | 0.181 | 0.208 | 0.121 |
| U10 | 0.205 | 0.186 | 0.091 | **0.081** | 0.217 | 0.185 | 0.224 | 0.112 |

**Table 2.** *Cont.*

| Scenario | S1 | S2 | S3 | S4 | S5 | S6 | S7 | S8 |
|----------|------|------|------|------|------|------|------|------|
| U11 | 0.092 | 0.108 | 0.046 | **0.042** | 0.195 | 0.116 | 0.115 | 0.058 |
| U12 | 0.200 | 0.127 | 0.067 | **0.047** | 0.123 | 0.111 | 0.158 | 0.097 |
| U13 | 0.101 | 0.103 | 0.051 | **0.048** | 0.112 | 0.077 | 0.103 | 0.056 |
| U14 | 0.195 | 0.201 | 0.115 | **0.072** | 0.238 | 0.163 | 0.188 | 0.127 |
| U15 | 0.101 | 0.148 | **0.045** | 0.049 | 0.190 | 0.157 | 0.157 | 0.092 |
| U16 | 0.152 | 0.152 | 0.107 | **0.100** | 0.147 | 0.127 | 0.172 | 0.127 |
| U17 | 0.034 | 0.156 | **0.020** | 0.077 | 0.183 | 0.120 | 0.153 | 0.121 |
| U18 | 0.256 | 0.207 | **0.052** | 0.066 | 0.220 | 0.075 | 0.172 | 0.097 |
| U19 | 0.199 | 0.205 | **0.105** | 0.113 | 0.280 | 0.149 | 0.221 | 0.134 |
| U20 | 0.117 | 0.130 | 0.077 | **0.073** | 0.220 | 0.149 | 0.169 | 0.103 |
| U21 | 0.025 | 0.028 | **0.016** | 0.037 | 0.027 | 0.029 | 0.026 | 0.027 |
| U22 | 0.115 | 0.139 | **0.092** | 0.111 | 0.134 | 0.111 | 0.136 | 0.132 |
| U23 | 0.274 | 0.190 | 0.064 | **0.055** | 0.191 | 0.147 | 0.164 | 0.077 |
| U24 | 0.136 | 0.126 | **0.107** | 0.125 | 0.128 | 0.154 | 0.121 | 0.123 |
| U25 | 0.052 | 0.111 | **0.042** | 0.056 | 0.357 | 0.134 | 0.175 | 0.063 |
| U26 | 0.157 | 0.168 | 0.110 | 0.116 | 0.175 | **0.098** | 0.172 | 0.152 |
| U27 | 0.195 | 0.207 | 0.221 | 0.238 | 0.337 | 0.159 | 0.183 | **0.158** |
| U28 | 0.128 | 0.130 | 0.093 | **0.077** | 0.136 | 0.128 | 0.132 | 0.108 |
| U29 | 0.093 | 0.166 | **0.039** | 0.080 | 0.194 | 0.153 | 0.156 | 0.093 |
| U30 | **0.070** | 0.178 | 0.092 | 0.154 | 0.182 | 0.279 | 0.129 | 0.085 |
| U31 | 0.012 | | **0.010** | | | | 0.038 | 0.034 |
| U32 | 0.143 | 0.143 | | | 0.167 | | **0.017** | |
| U33 | 0.271 | 0.280 | 0.128 | **0.093** | 0.349 | 0.129 | 0.226 | 0.108 |
| U34 | 0.148 | 0.153 | 0.091 | **0.081** | 0.175 | 0.116 | 0.175 | 0.111 |
| U35 | 0.066 | 0.182 | **0.034** | 0.097 | 0.287 | 0.238 | 0.212 | 0.114 |
| U36 | 0.108 | 0.181 | **0.064** | 0.126 | 0.200 | 0.176 | 0.217 | 0.152 |
| U37 | 0.084 | 0.138 | **0.043** | 0.076 | 0.233 | 0.145 | 0.199 | 0.068 |
| U38 | 0.118 | 0.167 | **0.082** | 0.140 | 0.167 | 0.158 | 0.155 | 0.148 |
| U39 | 0.109 | 0.111 | 0.070 | **0.067** | 0.151 | 0.133 | 0.140 | 0.108 |
| U40 | 0.083 | 0.139 | **0.048** | 0.059 | 0.203 | 0.188 | 0.324 | 0.094 |
| U41 | **0.096** | 0.141 | 0.104 | 0.161 | 0.156 | 0.206 | 0.160 | 0.134 |
| U42 | 0.142 | 0.158 | 0.101 | **0.085** | 0.161 | 0.188 | 0.161 | 0.114 |
| U43 | 0.283 | 0.363 | **0.051** | 0.067 | 0.450 | 0.219 | 0.135 | 0.066 |
| U44 | 0.043 | 0.067 | **0.039** | 0.076 | 0.076 | 0.074 | 0.078 | 0.067 |
| U45 | 0.158 | 0.164 | 0.095 | **0.064** | 0.222 | 0.177 | 0.165 | 0.079 |
| U46 | 0.117 | 0.142 | **0.043** | 0.088 | 0.303 | 0.138 | 0.155 | 0.111 |
| U47 | 0.134 | 0.133 | 0.091 | **0.065** | 0.168 | 0.130 | 0.153 | 0.095 |
| U48 | 0.166 | 0.229 | **0.121** | 0.129 | 0.377 | 0.208 | 0.210 | 0.129 |
| U49 | 0.086 | 0.176 | **0.064** | 0.133 | 0.181 | 0.168 | 0.178 | 0.137 |
| U50 | 0.148 | 0.144 | 0.111 | 0.115 | 0.148 | 0.140 | 0.142 | **0.109** |
| U51 | 0.109 | 0.142 | **0.105** | 0.114 | 0.222 | 0.198 | 0.217 | 0.127 |
| U52 | 0.109 | 0.216 | **0.102** | 0.140 | 0.304 | 0.184 | 0.217 | 0.148 |
| U53 | 0.053 | 0.153 | **0.035** | 0.108 | 0.162 | 0.152 | 0.152 | 0.145 |
| U54 | 0.119 | 0.099 | 0.051 | **0.037** | 0.294 | 0.124 | 0.086 | 0.048 |
| DP | 0.062 | 0.053 | 0.037 | 0.038 | 0.077 | 0.044 | 0.054 | 0.035 |
| Méd. | 0.135 | 0.162 | 0.076 | 0.088 | 0.208 | 0.151 | 0.163 | 0.102 |

Scenario 1, which applied the ARIMA/SARIMA models without any data treatment, presented an average MAE of 0.13 (lowest average among the scenarios of the ARIMA/SARIMA models) and a standard deviation of 0.06 (second highest standard deviation among all the scenarios). This result shows that, among the scenarios of the ARIMA models, the errors were smaller, on average. The variations indicate that the absence of data processing can reduce errors in specific scenarios or penalize the results. This scenario also obtained over 63% of the units with the lowest metrics among Scenarios 1, 2, 5, and 7, indicating that it prevailed in results with lower errors.

Scenario 2, which also applied the ARIMA/SARIMA models with outlier treatment, presented an average MAE metric of 0.16, with a standard deviation of 0.05, highlighting 11% of the units by the SARIMA model and 26% by the ARIMA model. Scenario 5, characterized by removing outliers, showed the highest error average and the highest standard deviation among all the scenarios. The same occurred for Scenario 6, which implemented the same data processing, but for the LSTM model, resulting in the highest error rates among the scenarios that implement this model. This indicates that removing points outside the curve impairs the predictions and does not improve the models' results. Scenario 7 uses sophisticated data processing, such as detecting anomalies before performing data processing. The metrics of this scenario are close to the metrics of Scenario 2, which also received data processing, resulting in an average MAE error of 0.16 and a standard deviation of 0.05.

Evaluating all the scenarios and units, the LSTM model presented better results. This can be observed in Figure 8, which presents representative samples of the overall results. Scenarios 3 and 4 represent 85% of the metrics among the units with the lowest errors when comparing all the scenarios, while the scenarios representing the ARIMA/SARIMA models represent only 6%. In addition, data processing did not contribute as expected to reduced errors. Anomaly detection methods did not contribute to error reduction, resulting in metrics that indicate more significant errors than conventional outlier treatment. Figure 9 illustrates the relationship between the forecast and actual values for all the scenarios.

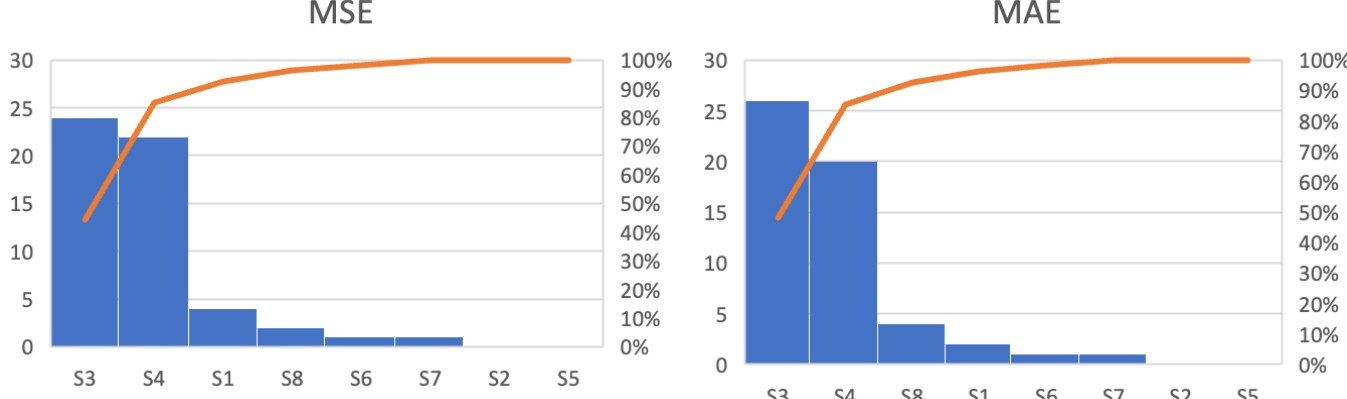

**Figure 8.** Aggregate number of shipping units with the smallest errors in each scenario with accumulated percentage curves, represented by the orange line.

The charts in Figure 10 summarize the metrics by counting the scenarios presenting the lowest average metrics. Scenarios 6 and 8 represented less than 12% of the total units evaluated, indicating that removing outliers from the dataset did not contribute to improved predictions. The treatment of outliers through sophisticated methods such as anomaly detection showed no reduction in error. Conversely, Scenarios 3 and 4 obtained over 88% of the highlighted metrics, suggesting that data manipulation can be advantageous in specific datasets.

For LSTM, the batch hyperparameter values of 32, 64, and 128 obtained equivalent proportions in scenarios/units with the highlighted metrics, obtaining values of 32%, 38%, and 30%, respectively. This result may indicate that this hyperparameter does not have a predominant value and can vary according to the dataset. In the case of epoch size, an increase in its values was observed to lead to increased errors. This may be related to the occurrence of overfitting. As for the lag size, the best value observed was 7 (due to the weekly demand's seasonality). Lastly, increasing the number of hidden layers was observed not to improve the quality of the results.

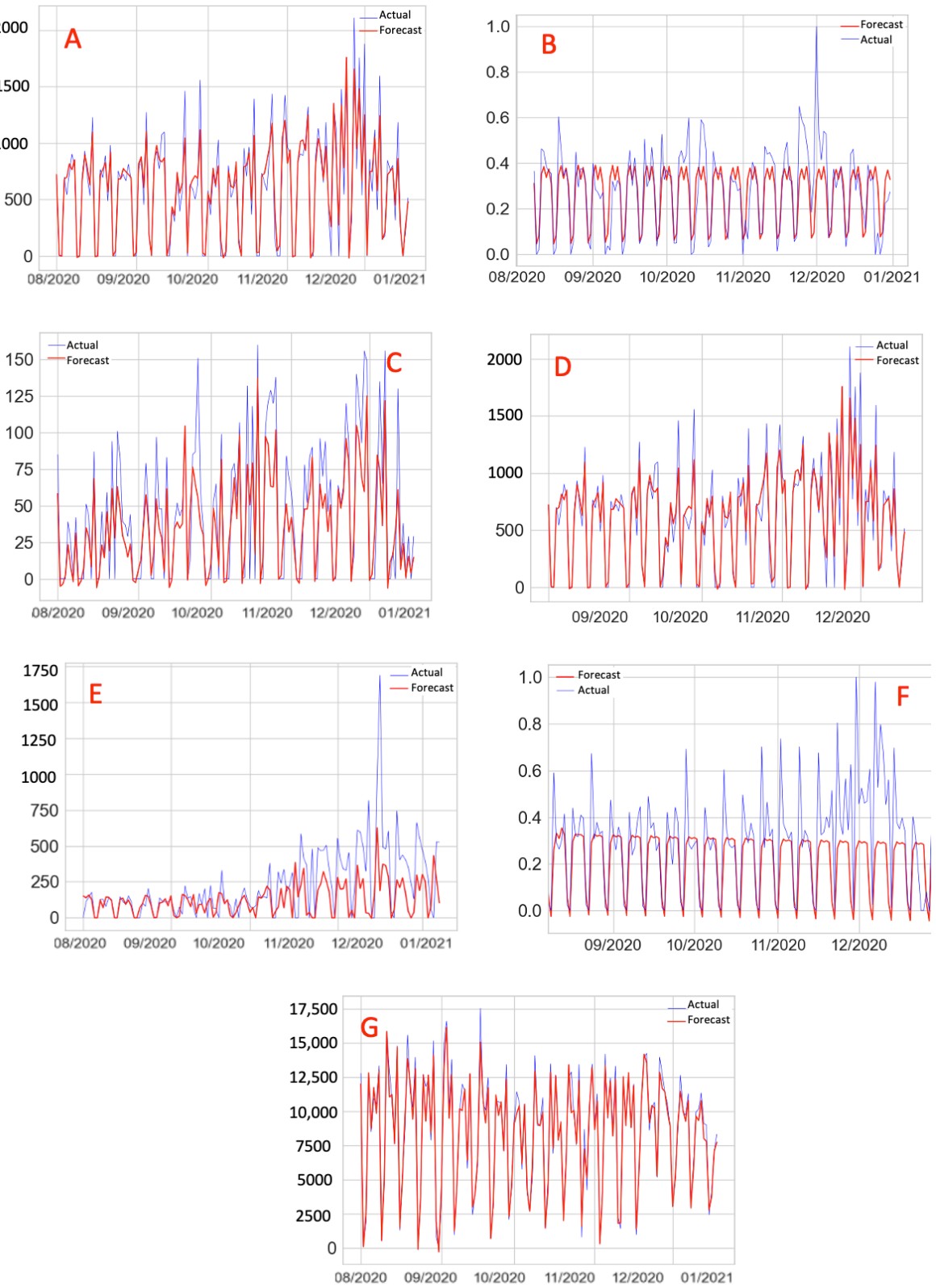

**Figure 9.** Forecast x Actual, Scenarios 1 (**A**), 2 (**B**), 3 (**C**), 4 (**D**), 6 (**E**), 7 (**F**), 8 (**G**).

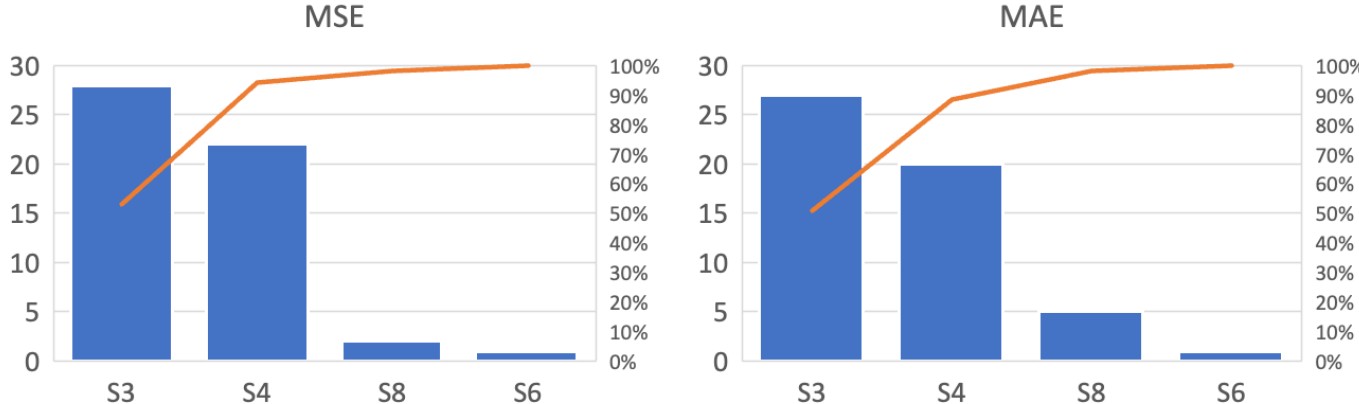

**Figure 10.** Aggregate number of shipping units with the smallest errors in each scenario for LSTM with accumulated percentage curves, represented by the orange line.

Table 3 presents the number of shipping units in scenarios that obtained the metrics with the lowest error for the number of neurons hyperparameter. The extremes (lowest and highest values) are observed to encompass 85% of the cases evaluated. This may indicate that the number of neurons in an LSTM polarizes the values that can present good results in extremes.

**Table 3.** Relationship between number of neurons, scenarios, and total units selected with the hyperparameter.

|  | S3 | S4 | S6 | S8 | Total |
|---|---|---|---|---|---|
| 16 | 4 | 2 | 1 | 0 | 7 |
| 32 | 6 | 2 | 0 | 1 | 9 |
| 64 | 0 | 2 | 0 | 0 | 2 |
| 128 | 1 | 4 | 0 | 1 | 6 |
| 256 | 6 | 2 | 0 | 2 | 10 |
| 1024 | 10 | 8 | 0 | 1 | 19 |

*4.3. Computational Performance*

Comparing the three models from the point of view of computation time, the LSTM model requires a higher processing time. This can be observed in Figure 11A,B, in which the minimums and maximums are referenced. The minima of the LSTM model vary from 5000 s, while the ARIMA/SARIMA models vary by about 300 s. The maximum times follow the same characteristic, with ARIMA and SARIMA not exceeding 700 s, while LSTM reaches nearly 80,000 s. These results are expected since the training process of determining neural network hyperparameters requires computational effort and processing complexity that overcome regressive models. Observing the minimums (Figure 11C) and maximums (Figure 11D) of the ARIMA/SARIMA models, the curves of the SARIMA model are verified to be below those of ARIMA. This is justified by the method for determining the hyperparameters of the SARIMA model, which uses heuristics with performance that surpasses the method of the ARIMA model.

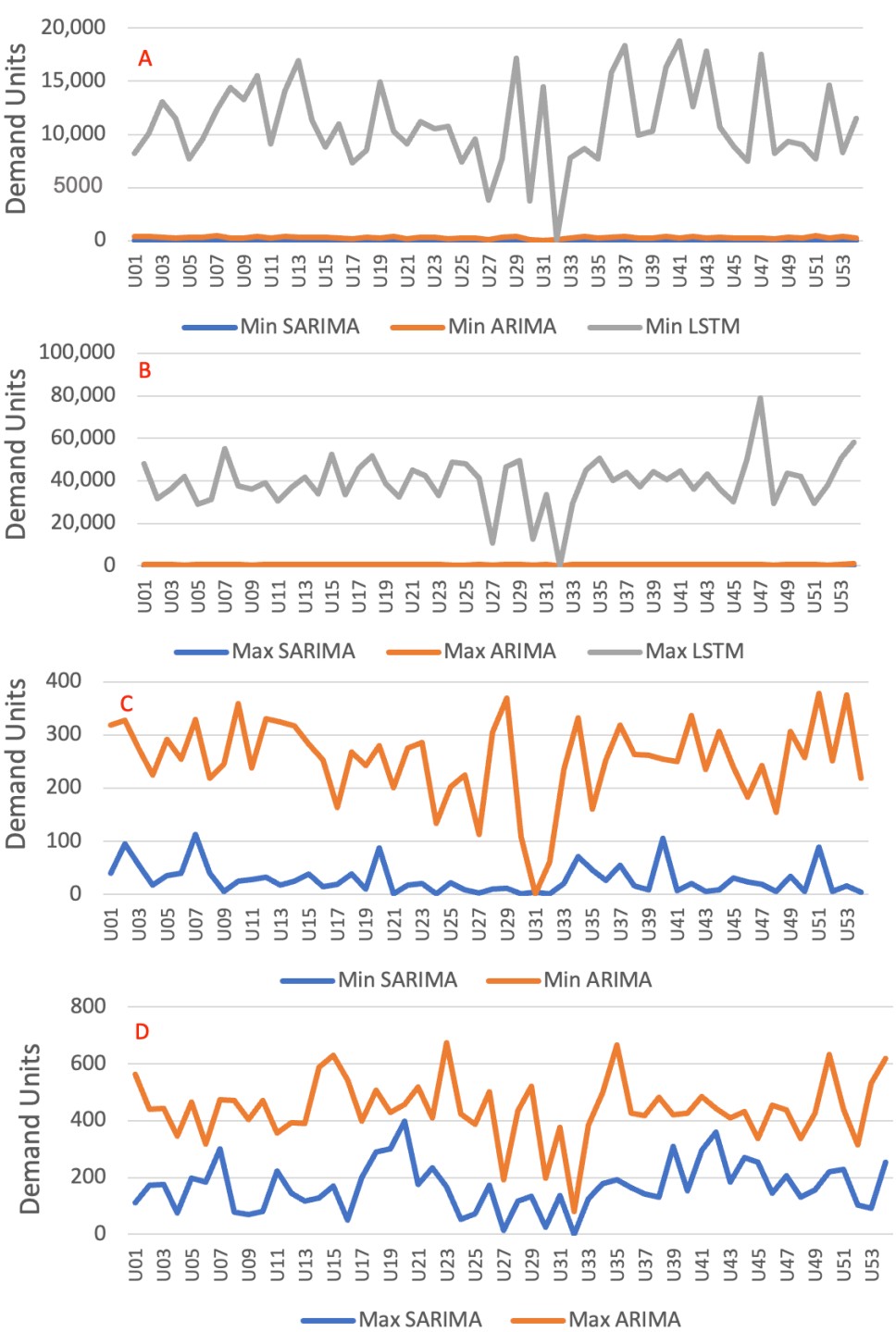

**Figure 11.** Minimum (**A**) and maximum (**B**) times between shipping units in all the scenarios in all the models. Minimum (**C**) and maximum (**D**) times between shipping units in all the scenarios in ARIMA models.

## 5. Discussions

Notice that 94% of the units evaluated obtained the lowest MAE error metrics when the LSTM model was used. These results align with state-of-the-art works in the literature, such as [6,9,12,13]. However, this work considers a broad range of datasets with different characteristics and transportation demands, adding important results to the literature.

A hypothesis explored in the literature is that neural networks can detect patterns and adapt to non-linear time series [5,7,23,33,39]. This allows methods such as the LSTM

model to outperform linear-based methods. Additionally, in the present case study, this model was able to identify patterns in datasets with very different characteristics, such as varying trends over the analyzed period, seasonality, and different impacts of the COVID-19 pandemic.

Considering an average error margin of 10% over 72 days, it can be concluded that the results could be satisfactory for operational decision-making arising from forecasts of transport demands. From the perspective of computational performance, LSTM requires more significant computational resources, with an average processing time of 167 to 88,000 times greater than that of the ARIMA/SARIMA models, depending on the scenario and the shipping unit. The training time average for the statistical models was 3.5 min, while for LSTM, it was 6.5 h. This difference is relevant for choosing models for real application situations.

Lastly, removing outliers was noticed not to reduce forecast errors. This was an interesting observation, as data processing was expected to reduce error metrics and allow the models implemented to be more effective. This could be further explored in future works, considering different datasets and data processing methods. The effects of the pandemic were evident in about half of the datasets and were concentrated between February and April 2020, with the results suggesting that the metrics affected were those of the ARIMA/SARIMA models. These models showed a lower capacity to adapt to contexts of fluctuations in relation to the LSTM model.

## 6. Conclusions

Forecasting transportation demands in different scenarios and situations is essential to improve decision-making in supply chain management. A gap in model comparisons was identified in the literature. To fill this gap, an in-depth case study was conducted considering two traditional models (ARIMA and SARIMA) and the state-of-the-art model (LSTM). Data from 54 units with different characteristics and demands were evaluated, considering error metrics and computational time.

The results showed that the outlier treatment did not contribute to a significant reduction in error metrics. The LSTM model achieved the best results in terms of error metrics. However, it required significantly more training time, making it unfit for real-time decisions that demand model retraining (for example, if an impactful event occurs and disrupts the supply chain, this model will require several hours for training). Therefore, it may be interesting to maintain both models in real-world scenarios, with the ARIMA and SARIMA models being used only when considerable disruptions occur and need a fast response.

Nevertheless, the predictions from the LSTM model have sufficient precision to be used for practical purposes, providing vital information to improve activities such as: (i) scheduling the number of vehicles that must be available on certain days and places and assertively repositioning them; (ii) scheduling labor related to load demands and drivers; (iii) vehicle purchase or sale planning; (iv) the dimensioning of shipping units; (v) billing forecasts; (vi) indicators and reports; and others. The first improvement mentioned (truck availability and repositioning) is the key implication and resolves issues such as resource planning in companies with numerous facilities spread across different locations. This problem is quite complex and needs an accurate forecast for decision making.

The primary limitations found in the research were: (i) the computational time demanded conducting the grid search method for the LSTM model; (ii) only the most critical hyperparameters were evaluated, but more can be explored in future works; (iii) difficulty was noted in obtaining further data; and (iv) the fact that all data referred to a single company. Future works are related to: (i) exploring additional artificial intelligence models, such as Bi-LSTM and Transformers, among others; (ii) evaluating other units and periods; (iii) adding variables that may help to identify outliers and supply chain disruption, such as market sentiment extracted from different sources; and (iv) evaluating additional unsupervised and supervised outlier detection and treatment methods, which can improve data quality.

**Author Contributions:** Conceptualization, F.P.M., R.F.d.S. and C.E.C.; methodology, F.P.M. and C.E.C.; software, F.P.M. and R.F.d.S.; validation, R.F.d.S. and C.E.C.; formal analysis, F.P.M., R.F.d.S. and C.E.C.; investigation, F.P.M.; writing—preparation of the original draft, F.P.M., R.F.d.S. and I.d.B.J., writing—revision and editing, F.P.M., R.F.d.S., I.d.B.J. and C.M.H.; supervision, H.T.Y.Y. and C.E.C.; obtaining funding, I.d.B.J. and H.T.Y.Y. All authors have read and accepted the published version of the manuscript.

**Funding:** This research received no external funding.

**Institutional Review Board Statement:** Not applicable.

**Data Availability Statement:** The data presented in this study are available on request from the corresponding author. The data are not publicly available due to privacy restrictions.

**Acknowledgments:** Irineu de Brito Junior Roberto Fray da Silva, and Hugo Tsugunobu Yoshida Yoshizaki are grateful to the National Council for Scientific and Technological Development (CNPq), 404803/2021-0, Brazil.

**Conflicts of Interest:** The authors declare no conflict of interest.

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
