# Peer review of "Deep Learning and Statistical Models for Forecasting Transportation Demand: A Case Study of Multiple Distribution Centers"

_logistics, 2023_

Round 1

Reviewer 1 Report

Comments and Suggestions for Authors

In the abstract, the methodology, findings, and implications must be explicitly informed;

Please strictly avoid lumped references. For example, in line 25, the uncertainties must be explicitly mentioned one by one. Otherwise, it would be less useful for other researchers, There are other all over the text, double check;

Although the English is fine, there are lots of typos, including observations in a language other than English (I suppose Portuguese). Please double check the entire text;

The research gap your study aims to bridge is not formally derived. Please lead a search in databases and theoretically evidence that there is a gap in the body of knowledge and that your study can bridge this gap. Furthermore, pose a research question, a formal purpose, secondary objectives and inform the research method and methodology in the Introduction.

The review is poor. Please delve more deeply in the main core of the study, the logistic problem. You reviewed only forecasts methods, which are intermediate steps. In the real world, the shop floor, what changes upon your study? Please review the state-of-the art of warehousing and crossdocking management;

The application is fine.

Please observe that the outliers treatment does not appear in Figure 3. Also I could not find how you detect outliers, please take a look, and explicitly inform.

Please observe Figures 4 and 7 and compare them with Figure 9. I believe you do not need to represent the std dev. Limiting the graphic to the current and predicted value should turn figures more readable.

Please consider merging small sections in order to have no third indentation level titles, as they turn the reading tedious and truncated;

In the last section, please insert concerns about the study's implications. Explictly answer who gains what and why upon your study.

You have few refs. You should rely on more refs for a top journal, employing only peer-reviewed articles from indexed journals (Scopus and WoS) issues from 2017 onwards. Please improve your set of refs.

Comments on the Quality of English Language

typos and parts in other language

Author Response

Reviewer number

Reviewers’ observation

Authors’ response

Original text

Modifications made on the text

Editor

E.1 Change personal email to institutional email

We agree with the suggestion, and changed the personal emails to institutional emails

fabiomamede50@gmail.com, roberto.fray.silva@gmail.com

fabiomamede@alumni.usp.br, roberto.fray.silva@usp.br

1, Editor

1.1. Standardize the format of the abstract by including the Background, Methods, Results, and Conclusions.

In the abstract, the methodology, findings, and implications must be explicitly informed

We fully agree with the suggestion. We have changed the abstract to include those topics in the right format

The transportation demand forecast is an essential activity for logistics operators and carriers. It leverages business operation decisions, infrastructure, management, and resource planning activities. Since 2015, there has been an increase in the use of deep learning models in this domain. However, there is a gap in works comparing traditional statistics and deep learning models for transportation demand forecasts. This work used actual operations data to perform a case study of aggregated transportation demand forecasts in 54 distribution centers of a Brazilian carrier. The computational simulation and case study methods were applied, exploring the characteristics of the datasets and evaluating three models: Autoregressive Integrated Moving Average (ARIMA) and its seasonal variation (SARIMA) and Long Short-Term Memory (LSTM) deep learning networks. Nine scenarios were explored, considering different data preprocessing methods and the role of outliers and exogenous variables on the demand forecasts. Moreover, different training and testing subsets split, and a grid search of the relevant hyperparameters for each model were evaluated. It was observed that the long short-term memory networks outperformed the ARIMA and SARIMA methods at 94\% of the dispatching units over the evaluated scenarios. The methodology and results can be applied to other scenarios, areas, and products.

Background: The transportation demand forecast is an essential activity for logistics operators and carriers. It leverages business operation decisions, infrastructure, management, and resource planning activities. Since 2015, there has been an increase in the use of deep learning models in this domain. However, there is a gap in works comparing traditional statistics and deep learning models for transportation demand forecasts. The objective of this work was to perform a case study of aggregated transportation demand forecasts in 54 distribution centers of a Brazilian carrier. Methods: The computational simulation and case study methods were applied, exploring the characteristics of the datasets, applied through Autoregressive Integrated Moving Average (ARIMA) and its variations and a deep neural network, Long Short-Term Memory, known as LSTM. Eight scenarios were explored, considering different data preprocessing, evaluating how the outliers can affect the demands forecasts. Moreover, splitting training and testing datasets during cross-validation, and relevant hyperparameters for each model. Results: It was observed that the long short-term memory networks outperformed the statistics methods in ninety four percent of the dispatching units over the evaluated scenarios, while the Autoregressive Integrated Moving Average models the remaining five percent. Conclusions: This work found that forecasting transportation demands can address practical issues in supply chain, specially resource planning management.

1

1.2. Please strictly avoid lumped references. For example, in line 25, the uncertainties must be explicitly mentioned one by one. Otherwise, it would be less useful for other researchers, There are other all over the text, double check

Ok, the references were split and detailed. Thanks for your comment.

1

1.3. Although the English is fine, there are lots of typos, including observations in a language other than English (I suppose Portuguese). Please double check the entire text

Ok, reviewed. Thanks for your comment.

1

1.4. The research gap your study aims to bridge is not formally derived. Please lead a search in databases and theoretically evidence that there is a gap in the body of knowledge and that your study can bridge this gap

Ok, reviewed

The literature has a few works with the context of transportation demand forecasts, however each one has a different focus, as logistics service capacity (Ren et al., 2020), business volume (Yuan et al., 2018), scheduling (Xi et al., 2020). None of these work and other references apply the transportation demand forecast in order to predict resource availability in facilities. Therefore, there is a gap in state-of-art to explore the business case of repositioning vehicles over facilities.

1

1.5. Furthermore, pose a research question, a formal purpose, secondary objectives and inform the research method and methodology in the Introduction

The paragraphs 5 and 7 of the Introduction section contains the objectives, secondary objectives and questions.

1

1.6. The review is poor. Please delve more deeply in the main core of the study, the logistic problem. You reviewed only forecasts methods, which are intermediate steps. In the real world, the shop floor, what changes upon your study? Please review the state-of-the art of warehousing and crossdocking management

A new subsection was created (2.1), extending the review and concepts about crossdocking and warehousing.

1

1.7. Please observe that the outliers treatment does not appear in Figure 3

It is described in the text "The sixth step identified potential outliers and different treatments were applied. Table 1 describes the 8 scenarios evaluated in this work."

1

1.8. Also I could not find how you detect outliers, please take a look, and explicitly inform

It is explained over Table 1 and the paragraph: "Scenarios 1 and 3 evaluate the ARIMA/SARIMA and LSTM models without any outlier treatment. Scenarios 2 and 4 evaluate ARIMA/SARIMA and LSTM performing outlier treatment, replacing the values identified by the lower or upper limit band. Scenarios 5 and 6 implement ARIMA/SARIMA and LSTM, discarding the identified outliers. Finally, Scenarios 7 and 8 implement ARIMA/SARIMA and LSTM with outlier treatment using advanced techniques for detecting anomalies in the data series. The Isolation Forest method from the Python library pycaret was used, replacing the points identified by the median of the data set. All scenarios considered all datasets and were built applying the cross-validation technique by dividing the training set into four parts (25\%, 50\%, 75\%, and 100\% of the total data in the training subset), as in the work by . Normalization was applied in all variables (transforming them in a range from 0 to 1) so that the model metrics were on the same scale.

1

1.9. Please observe Figures 4 and 7 and compare them with Figure 9. I believe you do not need to represent the std dev. Limiting the graphic to the current and predicted value should turn figures more readable

The graphic is the output of the program. To regenerate the graphic is going to reprocess the datasets and there is not enough time to reprocess until the submission deadline.

1

1.10. Please consider merging small sections in order to have no third indentation level titles, as they turn the reading tedious and truncated

Ok, removed sections 4.1.1 and 4.1.2 and merged with 4.1

1

1.11. In the last section, please insert concerns about the study's implications. Explictly answer who gains what and why upon your study

Ok, additional concerns included in the last section. Thanks for your comment.

1

1.12. You have few refs. You should rely on more refs for a top journal, employing only peer-reviewed articles from indexed journals (Scopus and WoS) issues from 2017 onwards. Please improve your set of refs

Ok, more than 10 new references added.

Reviewer 2 Report

Comments and Suggestions for Authors

This paper studies one of the most important aspect of supply chain logistics by brining in transportation demand forecasting using time series forecasting approaches. Three deep learning based methodologies were considered namely Autoregressive Integrated Moving Average (ARIMA), its seasonal variation (SARIMA) and Long Short-Term Memory (LSTM) networks.

The study consists of 2 phases, the computer simulation and case study. The computational model that allows the simulation of the characteristics of a specific problem, aiming to obtain insights and forecasts. In the second phase, model is implemented for actual data collected from a representative company in the market to distribute products of different natures, origins, and destinations.

In step 2 of the 9 step process, it would be essential to consider a timeframe along with the other variables such as weight, volume and origin unit.

Since feature dimension is low, I would suggest the authors to expand this study further to a larger and more granular data. Also considering annual volume, time and more unit locations. 

It would strengthen the paper further to introduce some more novelty into the existing LSTM and regression methodologies. Also, it is a good inclusion to add pandemic discrepancy, but it would be good to see the year 2021-2022, which was a challenge in the supply chain logistics area due to semiconductor shortage.

I would also suggest a more extensive literature review to cover the various DNN and RNN methods used in supply chain analytics.

Also, figure 1 referral on page 3 has a linking error.

Author Response

Reviewer number

Reviewers’ observation

Authors’ response

Original text

Modifications made on the text

2

2.1. In step 2 of the 9 step process, it would be essential to consider a timeframe along with the other variables such as weight, volume and origin unit.

Ok, reviewed. Thanks for your comment.

The variables collected were: (1) weight; (2) volume; and (3) origin unit.

The variables collected were: (1) weight; (2) volume;(3) origin unit and (4) demand date.

2

2.2. Since feature dimension is low, I would suggest the authors to expand this study further to a larger and more granular data. Also considering annual volume, time and more unit locations.

The data was provided by a logistics provider company (3PL), for a timeframe. Additional data is not available.

2

2.3. It would strengthen the paper further to introduce some more novelty into the existing LSTM and regression methodologies

The Introduction was reviewed to clarify the scientific gap and the data science focus of the paper.

The main goals expected for this work are: (1) exploratory data analysis, in order to 54
understand the data characteristics; (2) implement ARIMA and LSTM models.

2

2.4. Also, it is a good inclusion to add pandemic discrepancy, but it would be good to see the year 2021-2022, which was a challenge in the supply chain logistics area due to semiconductor shortage.

The data was provided by a logistics provider company, for a timeframe. Additional data is not available.

2

2.5. I would also suggest a more extensive literature review to cover the various DNN and RNN methods used in supply chain analytics.

Ok, subsection 2.3 now is covering additional details of Neural network methods.

2

2.6. Also, figure 1 referral on page 3 has a linking error.

Ok, error fixed. Thanks for your comment.

Reviewer 3 Report

Comments and Suggestions for Authors

Refer to the attached file, please.

Author Response

Reviewer number

Reviewers’ observation

Authors’ response

Original text

Modifications made on the text

3

3.1. The main goals of this study should be clearly stated, and some findings related to the main goals need to be addressed in result section

Ok, reviewed. Thanks for your comment.

The main goals expected for this work are: (1) exploratory data analysis, in order to understand the data characteristics; (2) implement ARIMA and LSTM models with its key hyperparameters and (3) evaluate results and verify the forecasts accuracy and processing time

3

3.2. Optimized parameters for ARIMA(P,Q,D), seasonal ARIMA(P,Q,D,p,q,d), and LSTM should be presented. AIC values for optimized conditions for ARIMA family models are also presented

There are 54 data sets, 8 scenarios and 3 methods (ARIMA,SARIMA, LSTM), that means the table would contain 1296 rows and 11 columns (P,Q,D,Ps,Qs,Ds,Neuron,batch,epoch, lagsize, hidden layers). We consider it is too large to present. If necessary, we can make it available as supplementary material.

3

3.3. Prediction horizon, i.e. step ahead, should be stated

The text describes that the prediction is one step ahead.

Round 2

Reviewer 1 Report

Comments and Suggestions for Authors

The authors addressed only partially the main issues.

Figures 4 and 7 are still difficult to read. Please remove the standard deviation, turning them consistent with Figure 9, more clear. Inadequate observations in Portuguese (I suppose) still remain, please double check the entire text.

Author Response

Reviewer number

Reviewers’ observation

Authors’ response

Original text

Modifications made on the text

1

Figures 4 and 7 are still difficult to read. Please remove the standard deviation, turning them consistent with Figure 9, more clear.

We reviewed Figures 4 and 7, removing the standard deviation.

N/A

Lines 281 and 305

1

Inadequate observations in Portuguese (I suppose) still remain, please double check the entire text

We reviewed the English and removed the text in Portuguese in the References section

N/A

Lines 441 to 543

Reviewer 2 Report

Comments and Suggestions for Authors

No further comments. 

Author Response

Reviewer number

Reviewers’ observation

Authors’ response

Original text

Modifications made on the text

2

No further comments.

Thanks for your comment.

N/A

N/A

Round 3

Reviewer 1 Report

Comments and Suggestions for Authors

Please refer to equation (1) in the text by … Equation (1)’ not by … the following equation

Author Response

Reviewer number

Reviewers’ observation

Authors’ response

Original text

Modifications made on the text

1

Please refer to equation (1) in the text by … Equation (1)’ not by … the following equation

Thanks for your comment. We reviewed Equation (1) according to your comment and we also reviewed the rest of the manuscript to check for errors elsewhere. 1 error found (in Figure 3) was also corrected.

…as shown in the following equation …

Line 124

… as shown in Equation 1: